# OpenReview forum: "Feasibility of Automatically Detecting Practice of Race-Based Medicine by Large Language Models"
_AAAI.org/2024/Spring_Symposium_Series/Clinical_FMs — AAAI 2024 SSS on Clinical FMs_

### Official Review · Reviewer_GZnZ · 2024-02-21
**Review of "Feasibility of Automatically Detecting Practice of Race-Based Medicine by Large Language Models" - Needs revision**

**Rating:** 6
**Confidence:** 3

**Review:**

# Peer Review for the Manuscript: "Evaluating Large Language Models for Race-Based Medical Content"

## General Evaluation

The paper presents an interesting study on the use of Large Language Models (LLMs) for identifying and evaluating race-based content in medical contexts. This topic is both timely and relevant, given the increasing reliance on LLMs across various sectors, including healthcare. The authors have made a commendable effort to highlight the challenges and nuances associated with race-based content in LLM outputs.

## Specific Feedback

### Strengths

1. **Relevance and Novelty:** The study addresses a critical gap in the current understanding and evaluation of LLMs in handling sensitive and crucial topics such as impact of racial stereotypes in medical advice.
2. **Methodological Approach:** The structured comparison across different LLMs provide a solid basis for the study's findings.

### Areas for Improvement

1. **Typos and Clarifications:** The paper mentions "nine unique LLM-prompt combinations" which should correctly be "twelve unique combinations." Attention to such details is crucial for the accuracy of the paper.
2. **Consideration of Skin Tone Variability:** The use of a more comprehensive skin tone classification, such as the Monk Skin Tone Scale ([Monk Skin Tone Scale](https://skintone.google/)), would enrich the study by providing a more nuanced understanding of race as it pertains to medical content. The current set of 13 prompts is limited, it is predominately representing black, white and some asian race. Recommend the authors to formulate a more diverse prompts by considering more examples of race-stereotype combinations.
3. **Benchmark Evaluation:** The paper states that there is a lack of methods to evaluate harmful content regarding race. This is not true, major players in Generative AI space like OpenAI, Meta, google all of them have released trust and safety scorecards and responsible AI covering bias and stereotypes is a big focus. However, existing benchmarks and datasets could be explored for their representation of race-related medical data. The absence of this exploration is a missed opportunity to contextualize the study's findings within the broader research landscape.
4. **Physician Backgrounds:** The background of physicians involved in the original research, particularly their awareness of bias and civil rights, is not detailed. This information is crucial for understanding the potential biases in the study's setup and interpretation.
5. **Statistical Measures:** A clearer explanation of the statistical measures used (Sensitivity, Specificity, NPV, PPV, F1) would make the paper accessible to a broader audience, including those not familiar with these terms.
6. **Methodology Suggestion:** Given the limitations of zero-shot prompting in niche domains like race-related medical data, exploring few-shot prompting or fine-tuning the models might yield more accurate results.

### Recommendations for Further Research

The authors are encouraged to explore the representation of race-related bias in publicly available datasets and benchmarks, such as those hosted on platforms like Hugging Face and Stanford's CRFM. Investigating these resources could provide insights into the current state of race representation in LLM training data and benchmarks. Furthermore, the authors should consider building and open-sourcing a dataset specifically for evaluating race-related content in medical advice. This contribution would significantly benefit the research community by providing a specialized resource for further studies.

## Some benchmarks for reference
- [Hugging Face Open LLM Leaderboard](https://huggingface.co/spaces/HuggingFaceH4/open_llm_leaderboard)
- [Stanford CRFM HELM Lite](https://crfm.stanford.edu/helm/lite/latest/)
- [Hugging Face Chatbot Arena Leaderboard](https://huggingface.co/spaces/lmsys/chatbot-arena-leaderboard)
- [Artificial Analysis](https://artificialanalysis.ai/)
- [Martian Leaderboard](https://leaderboard.withmartian.com/)
- [Hugging Face Enterprise Scenarios Leaderboard](https://huggingface.co/spaces/PatronusAI/enterprise_scenarios_leaderboard)

## Conclusion

The paper "Evaluating Large Language Models for Race-Based Medical Content" contributes important insights into the evaluation of LLMs for sensitive content. With the recommended revisions and further exploration of the highlighted areas, this paper has the potential to significantly impact the field.

---

### Official Review · Reviewer_LYU7 · 2024-02-22
**Detecting race-based LLM responses**

**Rating:** 6
**Confidence:** 4

**Review:**

The authors experimented with using LLM to detect whether LLM responses contain debunked race-based content. It's an important topic to explore and the results help us to better understand how current available LLMs may respond to race-based medical questions.

13 questions were used to generate the responses from LLMs. It's unclear whether the set of the questions are representative enough for the study. 11 out of the 13 questions contained direct mentions of races in the question themselves, while 2 questions were fairly general. There is no comparison between the two to see if there is significant difference. Since direct mentions of the races may be more likely to generate race-based responses, the result responses set may have significant higher race-based responses compared to real life clinical uses cases, which may bias the evaluation results.

The paper also didn't discuss on how the safe guard mechanism in proprietary and/or open source models may affect the model responses. In appendix, the author mentioned that MedPalm-2 simply rejects to answer some of the questions due to the content filter. This type of content filtering is a common practice and it may change constantly without any notice, especially for proprietary models, which puts a lot of uncertainty on the evaluation results.

---

### Official Review · Reviewer_8Ds8 · 2024-02-24
**This paper investigates the capability of GPT-3.5/4 as an evaluator to detect potential race-related bias in medicine. Overall this is a good work studying important problems in medicine.**

**Rating:** 7
**Confidence:** 4

**Review:**

Quality: The evaluation is comprehensive and convincing. The authors first chose the best evaluator agent using dataset from Omiye et al. Then the evaluator was used to assess candidate responses from 10 LLMs across 13 questions. The authors also studied different combinations of prompts, and showed GPT-4 with simple prompts is the best evaluator.

Clarity: The paper is well structured and easy to follow.

Significance: Race-based beliefs in healthcare can be harmful and it constructs a major concerns for doctors to apply LLMs in clinical settings. Conclusion from this paper is important to guide doctors to choose the best LLMs in practice.

---

### Official Review · Reviewer_Apz6 · 2024-02-26
**A simple paper with good evaluations of current SOTA models**

**Rating:** 5
**Confidence:** 4

**Review:**

An interesting benchmark  with good reproducibility.  Clear explanation and well written demonstrating how LLMS can be used to detect instances of race-based medicine.

Pros - Potentially an interesting read for a clinical audience who would like to know the feasibility of having such a model being run 'in clinical practise'.

Cons - This style of paper is very common (prompt crafting + evaluation) and can be criticised as not contributing anything novel to the field.